# Novel Photonic Applications of Silicon Carbide

**DOI:** 10.3390/ma16031014

**Published:** 2023-01-22

**Authors:** Haiyan Ou, Xiaodong Shi, Yaoqin Lu, Manuel Kollmuss, Johannes Steiner, Vincent Tabouret, Mikael Syväjärvi, Peter Wellmann, Didier Chaussende

**Affiliations:** 1Department of Electrical and Photonics Engineering, Technical University of Denmark, Ørsteds Plads, Building 343, 2800 Kongens Lyngby, Denmark; 2Crystal Growth Lab, Materials Department 6 (I-Meet), FAU Friedrich-Alexander University Erlangen-Nürnberg, Martensstr. 7, D-91058 Erlangen, Germany; 3Université Grenoble Alpes, CNRS, Grenoble INP, SIMaP, 38000 Grenoble, France; 4Alminica AB, Åsorp 2, 59053 Ulrika, Sweden

**Keywords:** silicon carbide, integrated photonics, material growth

## Abstract

Silicon carbide (SiC) is emerging rapidly in novel photonic applications thanks to its unique photonic properties facilitated by the advances of nanotechnologies such as nanofabrication and nanofilm transfer. This review paper will start with the introduction of exceptional optical properties of silicon carbide. Then, a key structure, i.e., silicon carbide on insulator stack (SiCOI), is discussed which lays solid fundament for tight light confinement and strong light-SiC interaction in high quality factor and low volume optical cavities. As examples, microring resonator, microdisk and photonic crystal cavities are summarized in terms of quality (*Q*) factor, volume and polytypes. A main challenge for SiC photonic application is complementary metal-oxide-semiconductor (CMOS) compatibility and low-loss material growth. The state-of-the-art SiC with different polytypes and growth methods are reviewed and a roadmap for the loss reduction is predicted for photonic applications. Combining the fact that SiC possesses many different color centers with the SiCOI platform, SiC is also deemed to be a very competitive platform for future quantum photonic integrated circuit applications. Its perspectives and potential impacts are included at the end of this review paper.

## 1. Introduction

Photonic integrated circuits (PIC), also named integrated lightwave circuits (ILC) or photonic lightwave circuits (PLC), have been discussed to overcome the predicted bottleneck for integrated circuits for more than half a century. During the development of PIC, a lot of different material platforms have been investigated, such as silicon (Si), silicon dioxide (SiO${}_{2}$), silicon nitride (SiN), aluminum gallium arsenide (AlGaAs), indium phosphide (InP), and the emerging gallium nitride (GaN), lithium niobate (LiNbO${}_{3}$), silicon carbide (SiC), etc. Although Si is playing the most important role in this area, benefiting from the well-established complementary metal-oxide-semiconductor (CMOS) processing, it is limited by its own optical properties, such as indirect bandgap, strong two-photon absorption at telecommunication wavelength and zero second-order nonlinearity. Therefore, other material platforms are developed either monolithically on their own or as hybrid platforms that are integrated with an Si substrate. Among them, SiC is emerging rapidly in this field, partially because of the commercially available high-quality crystal SiC wafers and mature micro and nanofabrication of SiC in the promotion of SiC powered electronic devices, a similar story as Si. Unlike many well defined figures of merit (FOM), such as Johnson FOM, Keyes FOM, Baliga FOM, and Baliga high-frequency FOM, for direct comparison of different material platforms in electronics, photonic material platforms are usually compared in terms of bandgap, refractive index, second-order nonlinearity, and third-order nonlinearity.

Normally, for an ideal PIC material platform, it has low loss at the working wavelength; has a high refractive index for strong light–matter interaction; has strong second-order nonlinearity for electro-optical modulation and second-harmonic generation; and has high third-order nonlinearity for efficient wavelength conversion. As shown in Table 1, SiC fulfills all the demanded optical properties: high refractive index, wide bandgap, and high second-order and third order nonlinearities. Additionally, SiC is CMOS compatible. Therefore, extensive investigations on SiC optical devices are emerging.

## 2. Silicon Carbide Photonic Integrated Platforms

Planar optical waveguides are key structures where the light is confined and transmitted by obeying total internal reflection (TIR). For SiC waveguides, it is indispensable to form an index contrast where SiC has a higher refractive index than the surroundings. At the beginning of this research, floating SiC in the air has been applied because the commercial SiC epilayers on Si substrates were available. Since Si has a higher refractive index than SiC, it has to be removed near the SiC waveguides. But this structure has restrictions on the post-etching process, such as patterning and top-cladding deposition. Instead, silicon carbide on insulator stacks is developed and applied in most of the publications currently. There are generally three different methods to form SiCOI, as shown in Figure 1. For method 1, smart/ion cut method is adopted from silicon-on-insulator (SOI) stack’s formation, where Si wafer is replaced by a SiC wafer [24,25]. But since SiC and Si have very different physical properties, the material loss in SiCOI is still quite high compared to SOI, after ion implantation. This method mainly delivers 4H SiCOI stacks. For method 2, the bonded SiC is thinned down directly by using grinding and later chemical mechanical polishing (CMP) to smoothen the surface. In this way, the ion implantation step in the smart/ion cut is skipped and SiCOI stacks achieve very low loss. This method delivers 4H and 3C SiCOI stacks. For method 3, amorphous SiC is directly deposited on the Si substrate with an SiO${}_{2}$ layer between. The amorphous SiC deposition could be completed through plasma-enhanced chemical vapor deposition (PECVD) and sputtering. Both methods are CMOS compatible.

Table 2 listed the state of the art of SiCOI waveguides in terms of waveguide structure, dimension, forming method, propagation loss, and quality factor (*Q*) of a microring resonator and the measured nonlinear refractive index. The active SiC layers have a thickness of hundreds of nm, which is extremely difficult to control in the process of thinning down SiC from hundreds of µm to nm. So far, 4H SiCOI waveguides formed from method 2 show the lowest loss of 0.38 dB/cm. 3C SiCOI from method 2 and amorphous SiC from method 3 have quite close losses of about 3 dB/cm. The loss of 4H SiCOI formed by method 1 could be derived from the quality factor of the microring, which is about 6.5 dB/cm. The waveguide loss is related to both the material loss and the waveguide fabrication imperfection. Nanofabrication technology has been well developed for other material platforms, such as SOI. SiC could be etched by using the same machine and precursors as Si. So, the fabrication-related loss is reduced to a negligible level for SiC in a very short time, while the supply of low-loss SiCOI stacks remains challenging. Comparing the listed 3 methods, method 2 shows the best performance, while method 1 and 3 have the similar performance, currently. However, with regard to the cost, method 2 is the most expensive one, followed by method 1 and 3 with reduced cost. In addition to the unique optical properties, SiC has the potential for easy scale-up. From this point of view, method 3 shows advantages because it is not limited by the rather small SiC wafers available, unlike method 1 and 2. Looking forward to quantum applications of SiCOI, many optically addressable defects in crystalline SiC such as 4H, 3C and 6H have been published [26,27], which are promising as candidates for single photon sources. But no report on optically addressable defects from amorphous SiC have been revealed yet. 6H SiCOI has been reported for usage in optical cavities [28,29,30,31,32]. It is used less, as 4H SiC is becoming dominant because of its better crystal quality, which is crucial for high quality-factor optical cavities. Moreover, many physical and optical properties of 6H SiC lie between 3C and 4H SiC, making it less interesting.

## 3. Silicon Carbide Photonic Cavities

The light-matter interaction could be enhanced by an optical cavity. Normally, a higher quality factor and a smaller mode volume are preferred. As an example for a quantitative relation, power threshold for parametric oscillation is determined by the waveguide loss (described by the loaded quality factors $Q_{L,s}$, $Q_{L,i}$, and $Q_{L,p}$ of the signal, idler, and pump modes, respectively) and the confinement, given by
(1)$$P_{th}=\frac{\omega_{0}n^{2}}{8\eta n_{2}c}\frac{V}{Q_{L,p}\sqrt{Q_{L,s}Q_{L,i}}}$$
where *n* is the modal refractive index, *V* is the mode volume, and η= $\frac{Q_{L,p}}{Q_{c,p}}$, where $Q_{c,p}$ accounts for coupling from the pump mode to the waveguide, $n_{2}$ is the nonlinear refractive index and $\omega_{0}$ is the angular frequency [38]. For SiC, three types of optical cavities have been published: photonic crystal cavities (PhC), microring resonators (MRR) and microdisk resonators (MDR). Their evolvement during the past ten years is shown in Figure 2.

Among the different forms of SiC (3C, 4H and 6H polytypes, and amorphous), there has been no further progress for 6H SiC in recent years. 4H SiCOI microring resonators made by method 2 have demonstrated a quality factor of more than 1 million. Based on it, advanced applications such as optical frequency combs have been demonstrated as well.

### 3.1. Microring and Microdisk Resonators

Thanks to the advanced nanofabrication technologies, SiC microresonators, including MRR and MDR, with low loss and high quality factors, have been demonstrated. The loss is jointly contributed by the material absorption loss and the light scattering loss due to the surface roughness. Figure 3 shows scanning electron microscope (SEM) images of several high-performance SiC microresonators.

For 3C SiC, using the freestanding SiC, grown on the Si substrate, quality factors of 2.4 × $10^{4}$ and 5.12 × $10^{4}$ have been reported for MRR and MDR, respectively [42,49]. The high loss mainly comes from the material absorption and is associated to the high density of extended defects, inherently linked to the 3C-SiC/Si heteroepitaxial system, and discussed in the coming section. The highly defective layer, which is close to the 3C-SiC/Si interface, can be removed, if preparing 3C SiCOI stacks. As a result, microresonators with much higher quality factors can be achieved, which are 1.42 × $10^{5}$ and 2.42 × $10^{5}$, for MRR and MDR, respectively [34,53]. For 4H SiC thin film prepared through the smart/ion cut method, quality factors of 7.3 × $10^{4}$ and 5.25 × $10^{3}$ have been reported for MRR and MDR, respectively [36,60]. Chemical-mechanical polishing and post-oxidation processes can also help to reduce the roughness of the top surface and the sidewall of the waveguides, and thus improve the quality factors of SiC microresonators, for both 3C and 4H SiC [34,36,61,62]. For 4H SiC thin film prepared through the grinding and CMP method, ultra-high quality factors of 1.1 × $10^{6}$ and 6.75 × $10^{6}$ have been reported for MRR and MDR, respectively [38,56]. Regarding the amorphous SiC deposited by PECVD, the MRR with a quality factor as high as 1.6 × $10^{5}$ has been reported [35]. Based on these demonstrations of high-performance SiC microresonators, as well as other SiC integrated devices, SiC integrated platforms can be potentially applied to classical and nonclassical optical communication systems for signal processing, logic gate, modulation, filtering, (de)multiplexing, etc. [5,24,63,64,65,66,67,68].

The field enhancement in the microresonators can be reflected by the photon density. With SiC microresonators, efficient nonlinear effects-based frequency conversion has been observed and studied, according to the material-based second- and third-order nonlinearities in SiC. Second harmonic generation in 4H SiCOI MRRs has been demonstrated with sub-milliwatt power, through the mode-phase-matching method [52]. Besides, basic third-order nonlinear phenomena, such as self-phase modulation, four-wave mixing, cascaded four-wave mixing, and optical parametric oscillation have also been demonstrated in 3C, 4H, and amorphous SiC or SiCOI microresonators [33,35,37,38,55,62,69,70]. Through these experiments, the nonlinear refractive index of 3C and amorphous SiC has been extracted to be 5.3 × $10^{-19}$ m${}^{2}/$W, and 4.8 × $10^{-18}$ m${}^{2}/$W, respectively [33,55]. The birefringence of the third-order nonlinearity of 4H SiC has been experimentally demonstrated, the nonlinear refractive index along the extraordinary and the ordinary axis of 4H SiC is measured to be 1.31 × $10^{-18}$ m${}^{2}/$W and 7.0 × $10^{-19}$ m${}^{2}/$W, respectively, which indicates the transverse magnetic polarization can perform more efficient wavelength conversion [23]. Moreover, classical and quantum frequency combs in 4H SiCOI MMR and MDR with ultra-high quality factors are observed, including octave-spanning Kerr frequency combs, Raman combs, and quantum solitons, which are shown in Figure 4 [56,57,71]. The excellent optical nonlinear properties of SiC, as well as these nonlinear phenomenon demonstrations, make SiC integrated platforms possible for plenty of nonlinear optical applications, such as light sources with multiple wavelength for optical communication, photon-pair sources for quantum computing and communication, spectroscopy, metrology, sensing, etc. [3,4,72].

Additionally, the thermo-optic behaviors of the MRRs in 3C, 4H and amorphous SiC integrated platforms have been studied. The external temperature induced resonance shift and the material absorption-induced bistability in 4H and amorphous SiCOI are investigated [55,58,62]. Micro-heaters have been integrated with the 3C SiCOI MRR for efficient thermo-optic phase shifting [51]. Meanwhile, the thermo-optic coefficients of 3C, 4H and amorphous SiC are measured to be 2.67 × $10^{-5}$/K, 4.21 × $10^{-5}$/K, and 1.4 × $10^{-4}$/K, respectively [51,55,62]. Devices made of 3C and 4H SiC that have small thermo-optic coefficients, can exhibit high thermal stability, but amorphous SiC, having a large thermo-optic coefficient, allows efficient modulation. The electro-optic effect in 3C SiCOI has also been observed, and the Pockels coefficient is extracted to be 2.6 pm/V, which enables high-speed electro-optic modulation [73,74]. An all-optical Kerr switching intensity modulator at 25 Gbps in amorphous SiCOI has been reported [75].

### 3.2. Photonic Crystal Cavities

Besides the microring and microdisk, the cavities realized by photonic crystal structures are also widely studied in the SiCOI platform. Because of the potential of emitting single photons by defects in SiC, photonic crystal cavities are studied to enhance the emission rate of the defects by addressing the defects in the cavities. The amplified magnitude of the modified emission rate comparing with the defects in free space is expressed by the Purcell factor in Equation (Equation 2) [76]. In this equation, $F_{p}$ is the Purcell factor, *Q* is the quality factor of a certain cavity mode, $V_{0}$ is the mode volume, and λ is the wavelength of light transmitting in the dielectric material.
(2)$$F_{p}=\frac{3Q\lambda^{3}}{4\pi^{2}V_{0}}$$

In order to realize a large increase of the emission rate of photons from defects in SiC, the strong light–matter interaction is required. From Equation (Equation 2), the quality factor of the photonic crystal cavity should be as large as possible, and the mode volume is preferred to be kept small. Due to the large scattering loss, the quality factor of the PhC cavity is much smaller than that in the microring or microdisk.

The photonic crystal can be briefly classified into one-dimensional (1D) photonic crystal and two-dimensional (2D) photonic crystal in SiCOI platform. A 1D photonic crystal waveguide is also called a photonic crystal nanobeam. As long as the holes are modified in terms of positions or radius, a nanocavity will form and can support several modes in the nanobeam. A 2D photonic crystal can further be transformed into a 2D photonic crystal cavity by removing several holes in the center, such as one hole (H1 cavity) or three holes (L3 cavity), or a 2D photonic crystal waveguide by removing a line of holes. By tuning the lattice constant along the photonic crystal waveguide direction in several periods, a heterostructure cavity will be formed [77]. Regarding the application of color center photoluminescence enhancement, a 1D nanobeam is preferred to 2D photonic crystal structures because of its lower modal volume and the feasibility of the undercut to have a larger refractive index contrast.

For 6H SiC prepared by smart-cut, the L3 cavity and heterostructure cavity were firstly fabricated in a smart-cut SiCOI platform [28]. The quality factor of an optimized L3 cavity of $2\times 10^{3}$ and the quality factor of the heterostructure cavity of $4.5\times 10^{3}$ have been reported. These two kinds of cavities on 6H SiCOI were improved to $1\times 10^{4}$ quickly afterwards [29,30,31]. A second harmonic generation has been realized in a photonic crystal heterostructure cavity in the same platform [32]. For 3C SiC, the L3 cavities were realized with an initial quality factor of around $0.8\times 10^{3}$ in an epitaxially grown thin film [41]. The more detailed results have been reported with the quality factor of $1.5\times 10^{3}$ for L3 and around $1\times 10^{3}$ for H1, despite a theoretical prediction of $1.7\times 10^{4}$ and $4.5\times 10^{4}$, respectively [43]. The imperfection of fabrication and deviation of the hole radius or positions are likely to cause a low quality factor. For amorphous SiC, the highest reported experimental quality factor of 1D photonic crystal cavity is $7.69\times 10^{4}$, and the mode volume is around $0.6\times {\left(\lambda /n\right)}^{3}$ [45]. The Purcell factor is calculated to be around $10^{4}$. This cavity is a suspended 1D photonic crystal nanobeam cavity. The decline of experimentally measured quality factor, compared with a simulation result of $1.75\times 10^{6}$, is mainly from the scattering of imperfect sidewalls. For 4H SiC, the high experimental quality factor of 6.3 $\times 10^{5}$ is achieved in a ground SiCOI platform in a heterostructure cavity [50]. A photonic crystal nanocavity with such a high quality factor enables strong second-harmonic generation conversion efficiency. The scheme of the structure and the SEM images are shown in Figure 5b. In a 1D photonic crystal nanobeam cavity, the quality factor is smaller: $1.93\times 10^{4}$ [52]. The color centers inside the nanobeam cavity can enhance the emission efficiency; the structure is shown in Figure 5a. All these works point out the great reduction of experimental quality factor as compared with simulated or calculated quality factor. The non-vertical sidewall angle is the main cause of this difference. The other attributions include the sidewall roughness, deviation of hole radius or position, and material absorption.

## 4. Silicon Carbide Materials

Different from the electronic devices, the material requirements for SiC photonic applications handle surface roughness, film thickness and uniformity in nanometer scale, which poses big challenges on the material quality, transfer and processing. As mentioned in Section 2, 4H, 3C and amorphous SiC have shown loss at different levels. If we assume the nanofabrication has been optimized and the fabrication-related loss is negligible, the loss level indeed reflects the material loss. The material loss consists of absorption and the scattering of defects in the material. The most promising large-area thin-film-deposition processes (amorphous and micro-crystalline material) are of particular interest due to their huge variability in terms of stacking thin optical films on almost any kind of substrate and because of their comparably low manufacturing costs. The currently mid-range optical loss may be reduced by further improvement of the deposition processes. Therefore, in the following an overview on all SiC-based thin film methods currently under development will be given. The lowest optical loss may be achieved in single-crystalline SiC material exhibiting a dislocation density that is as low as possible, as well as unintentional doping. At the same time, by adaption of the single-crystal growth process, intrinsic point defects may be generated for single-photon-source or qubit applications. Besides the preparation of high-quality 4H-SiC crystals and wafers, also bulk growth of 3C-SiC will be outlined. The latter material exhibits a higher defect density than 4H-SiC. However, its higher crystalline symmetry may be of advantage at a certain stage if optical active centers (point defects and intentional doping) are added as a functionality to the electrical-optical devices.

### 4.1. Amorphous Silicon Carbide

Contrary to the SiC polytypes, which have a well-defined crystalline structure and composition, amorphous Silicon Carbide (a-SiC) is a very versatile material. It can vary in stoichiometry, such as the Si:C ratio, in density, and of course in the resulting optical, electronic and mechanical properties. Depending on the deposition method, it can also contain hydrogen (H:SiC), which provides even further tuning of the properties. At the first order, the primary parameter which determines the formation of amorphous SiC is the temperature, as it enhances the relaxation of the “structure” towards crystalline materials. There is thus a temperature threshold above which crystallization will systematically occur. This temperature is not precisely determined in the literature as it also depends on secondary parameters like supersaturation, reaction pathway, presence of impurities and so on. It is nevertheless strictly inferior to 800 °C. We review in the following the low temperature, vapour-phase deposition methods that allow the formation of amorphous SiC thin films.

#### 4.1.1. Low Temperature Chemical Vapour Deposition

A first process which can lead to amorphous SiC is the Low Temperature Chemical Vapour Deposition. Using standard chemistry (Si-C-H or Si-C-H-Cl), such as is discussed in the next section for epitaxial growth, it is not possible to deposit in the targeted temperature range, as the gaseous species are too stable. Differently said, the reaction kinetics are infinitely slow. However, with organometallic precursors such as monomethylsilane (MMS), a deposition temperature lower than 650 °C and down to room temperature can form a-SiC [78,79,80].

Contrary to the standard CVD, where the initial cracking of the precursors in the gas phase provides reactive intermediates, the MMS is directly adsorbed on the surface of the sample. For that, the surface reactivity is promoted either by the deposition of a hydrogenated silicon layer or by a plasma etching [81], which enhances the adsorption efficiency. The hydrogen atoms get desorbed during this reaction. However, because of the low temperature of the process and the needed hydrogen desorption, the stoichiometry of the SiC becomes hard to control [80,81]. No optical study using films deposited by this process was found in the literature.

#### 4.1.2. Plasma-Enhanced Chemical Vapour Deposition

Plasma-Enhanced CVD (PECVD) is probably the most widely used technique for the deposition of a-SiC thin films [82,83]. The substrate is kept at a temperature lower than 500 °C, and often even lower than 300 °C. The chemical reactions are induced in a plasma (usually in Argon and/or hydrogen), which drastically increases the reactivity of the intermediate gaseous species. The ionic species produced are then adsorbed on the surface, whereas the organic wastes are evacuated. The energy of the ion bombardment allows the migration of the adsorbed species and the formation of dense layers. The use of hydrogen as carrier gas systematically leads to the hydrogenation of the SiC films. This limits the maximum refractive index achievable, which is typically around 2.36 [83]. The refractive index decreases with the density, which can be tuned by varying the plasma power [83,84,85,86], as shown in Figure 6.

An increase of the substrate temperature and of the pressure cause an increase of the refractive index [85,86] and the decrease of the roughness down to 1 nm [86].

#### 4.1.3. Hot Wire Chemical Vapour Deposition

Known as one of the pioneer CVD methods, Hot Wire CVD (HWCVD), also called Hot Filament CVD (HFCVD) is also well adapted for the deposition of amorphous SiC [87,88,89]. In this case, the energy necessary to crack the precursors and generate highly reactive intermediates is provided by a metallic filament composed of Rhenium, Tantalum or Tungsten placed upstream in the reaction chamber and heated at high temperature, typically around 2000 °C [90,91]. The thermally activated species are then adsorbed on the substrate, which can be kept at a low temperature, even lower than 500 °C. The presence of hydrogen radicals (H*) generated at high temperature by the filament produces an etching reaction of the surface, which can remove the adsorbed species located in unstable positions. A high concentration of H* can thus produce crystalline layers, with the possibility to tune the crystallite size with the H* concentration [87,92,93,94]. The formation of a stoichiometric, crystalline SiC layer is ensured by decomposing the carbon-containing precursor (typically CH${}_{4}$) at a temperature higher than 2000 °C [93]. At a temperature lower than 2000 °C, with low pressure and a low amount of hydrogen, the formation of an amorphous SiC layer can be obtained. Similarly to PECVD, this technique causes the incorporation of hydrogen inside the amorphous SiC film, decreasing the refractive index. The roughness of the SiC deposit can be also very low, in the nm range [95].

#### 4.1.4. Radio Frequency Magnetron Sputtering

Amorphous SiC can be deposited with a magnetron sputtering method. In this process, one or several targets with the composition aimed are eroded with a plasma. Then, the ionized species formed are adsorbed on the surface of the sample [96,97,98,99,100]. Non-conductive compounds used as targets, such as SiC or Si, require the use of Radio Frequency Magnetron Sputtering to prevent the accumulation of electric charge [101]. Amorphous layers can be deposited with a temperature lower than 500 °C [102,103]. As a rule, a low deposition temperature increases the refractive index up to 3.2 at a wavelentgh of 630 nm [103,104]; whereas it diminishes the transmittance of amorphous SiC [104], as shown in Figure 7.

Moreover, lowering the electrical power applied on the target increases the transmittance and diminishes the concentration of carbon clusters [103,105], as shown in Figure 8. Tang reported that decreasing the power diminishes the size of the particles and modifies their shapes from a dense globular-shaped structure to columns. This change of particle morphology diminishes the roughness [106]. The optical gap is evaluated at 3.5 eV [105,107], but the presence of graphite in the amorphous SiC can put the gap between 1.3 to 1.7 eV [103,105,108].

### 4.2. 3C Silicon Carbide

#### 4.2.1. Growth of 3C-SiC-on-Si Heteroepitaxy

Compared to its hexagonal counterparts (4H- and 6H-SiC) for the cubic polytype (3C-SiC), bulk growth using Physical vapour transport (PVT) is not established as “standard” method yet, which is mainly attributed to the lack of suitable seeding substrates. However, 3C-SiC is the only polytype that can be nucleated using heteroepitaxial chemical vapour deposition (CVD) on Si. Moreover, due to the higher process control available in CVD, the fine tuning of material properties especially the doping concentration is increased compared to PVT systems, especially for the goal of low-doped material. While silicon as a starting material can lower the cost for device fabrication due to the availability of large area substrates, a series of challenges have to be solved. The two main obstacles to overcome are the difference in the lattice constant of approx. 20% and the thermal expansion coefficient (TEC) of ca. 8% at room temperature between 3C-SiC and Si [109,110]. These misfits are responsible for a variety of defects formed at the interface, e.g., stacking faults (SFs), anti-phase boundaries (APB) or even three-dimensional defects such as protrusions [111] and additionally result in wafer-bending after the cool-down to room temperature [112]. The two most common defects in heteroepitaxial growth of 3C-SiC on Si are SFs and APB. SF along the {111} planes are mostly described as an intrinsic defect, based on the lattice mismatch and difference in TEC between 3C-SiC and Si. However, a decrease in the SF density could be achieved by annihilation during growth of thicker 3C layers [113] or advanced growth methods, such as the “switch back” epitaxy on undulated substrates described by Nagasawa et al. [114]. Different APBs are 2D defects that are formed by two 3C-SiC crystals, which are rotated 180° around the [110] axes. While getting rid of SF completely is somehow still impossible, the application of an off-axis Si substrate can lead to an APB-free material, as described in [115]. Today, the “standard” heteroepitaxial CVD growth process consists of a hydrogen etching step to remove the native oxide layer from the Si substrate, the growth of a buffer layer or carbonization layer and the main growth step, using Si and carbon (C) precursors followed by the cool down of the system. Traditionally silane (SiH${}_{4}$) and propane (C${}_{3}$H${}_{8}$) or ethylene (C${}_{2}$H${}_{4}$) are used while hydrogen (H${}_{2}$) acts as the carrier gas. However, a variety of different approaches can be found in the literature using different precursors or adding additional gas species. Often chlorine is added either in the form of hydrogen chlorine (HCl) or chlorine-based precursors like trichlorosilane (HCl${}_{3}$Si) to improve the material quality and increase the growth rate [116]. One of the most important aspects regardless of the used gases is the C/Si ratio during the main growth step. Tuning this parameter has not only an influence on the growth rate but also strongly influences the material quality. Chassagne et al. showed that for too-low C/Si ratios, the layer morphology and crystalline quality quickly degrade [117]. Meanwhile for too high ratios, homogenous gas phase reactions can occur leading to SiC clusters on the surface. Besides the correct adjustment of the C/Si ratio, the formation of the mentioned buffer layer is crucial when it comes to material quality. The first vital studies on this topic were conducted by Nishino et al. [118] and later by Liaw and Davis [112]. Both groups described the importance of a buffer layer formed by introducing a carbon-containing precursor prior to the main growth step during the heat up of the system. During this step the gaseous carbon species will produce an initial layer of SiC by reacting with the Si of the substrate. Due to the formation of this buffer layer the lattice mismatch between 3C-SiC and Si can be reduced significantly, leading to lower defect densities and increased material quality. A setback of the buffer layer is the formation of voids at the interface 3C-SiC/Si. Several theories can be found in the literature explaining this phenomenon. Most likely, the formation of voids is attributed to the surface diffusion of Si [119]. After the initial 3C-SiC seeds are formed due to the deposited carbon, Si is removed from the open substrate surface between these seeds to supply the formation of 3C-SiC, resulting in the observable voids (see Figure 9). Since voids can locally favor the formation of dislocations and SFs, a lot of research has been conducted to date to improve the buffer layer quality and reduce void formation. Possible approaches include the variation of the C/H${}_{2}$ ratio [120,121], the process pressure [122] or the initiation of Si precursors flow during the formation of the buffer layer [123].

Based on the lattice mismatch between 3C-SiC and Si, additional focus has to be given to the management of thermal stress induced by the difference of TECs between Si and 3C-SiC, leading to wafer-bending after the cool down from growth temperature. Zielinski et al. [109] also showed that the TEC mismatch cannot be changed; the final wafer-bow of heteroepitaxial grown 3C on Si can be controlled based on the C/Si ratio, final layer thickness or the growth rate during the deposition. Another approach was introduced by Anzalone et. al by in situ melting of the silicon substrate once a certain 3C thickness was reached, eliminating the influence of TEC during the cool-down [124,125]. After the silicon removal, it is also possible to further increase the growth temperature, enabling growth of a few 100 µm-thick freestanding 3C-SiC wafers with reasonable growth rates and diameters up to 150 mm. While the 3C-SiC heteroepitaxy on Si is mostly performed above 1300 °C and near to the melting point of Si [126], low temperature CVD between 1000 °C and 1200 °C represents a different approach for the growth of 3C SiC on Si [127,128]. To compensate for the lower temperatures resulting in decreased growth rates, chlorine is often added either in the form of HCl or chlorine-containing precursors [129]. Although the quality of the achieved layers is not the same as for high temperature processes yet, the low temperatures open up the route for the integration into existing CMOS fabrication technology.

#### 4.2.2. Heteroepitaxial Deposition on Hexagonal Substrates 4H SiC (CVD and PVT)

Additionally to CVD heteroepitaxy on Si, cubic SiC can be grown heteroepitaxially on hexagonal SiC substrates in either a CVD or physical vapour transport (PVT) setup. Using SiC as starting material enables higher growth temperature of up to 2000 °C and reduces the lattice mismatch as well as the difference in thermal expansion between substrate and epitaxial layer. For both cases (CVD and PVT), a (0001) orientated SiC substrate is usually used as a starting point, which will result in a (111) orientated 3C SiC top layer by “pseudomorph growth” [130]. During homoepitaxial growth of hexagonal SiC polytypes “step-controlled epitaxy” is preferred to ensure stable reproduction of the substrate polytype during growth [131,132,133]. Conditions favoring this growth regime are a low supersaturation at the growth surface as well as high off-axis angles (3–8${}^{\circ }$) resulting in a high number of atomic growth steps. The high step density and small terrace width, respectively, lead to a migration of adatoms towards step edges where they reproduce the substrate polytype during incorporation into the crystal. For the nucleation of 3C SiC on the hexagonal substrate, the growth conditions have to be tweaked from the nominal step flow towards the formation of 2D nucleation on the growth terraces. This growth regime can be promoted if the probability of adatoms reaching a growth step is reduced by either lowering of the growth temperature, increasing the terrace width by growth on “on-axis” substrates or increased supersaturation. A schematic scheme of the two growth modes are depicted in Figure 10.

Various studies can be found in the literature [134,135,136,137,138] working on the heteroepitaxial growth of 3C SiC on (0001) orientated hexagonal SiC up to a thickness of 2.5 mm using the sublimation epitaxy (SE) growth technique. They used a different off-axis substrate, on which an on axis facet will be formed at the initial growth stage, acting as a preferential nucleation side for the cubic polytype. Starting from this side, the 3C will laterally expand due to step flow growth covering the hexagonal substrate completely. The thickness necessary for a complete coverage is thereby predetermined by the use of the cut angle of the substrate and will increase for higher off-cuts. The SE setup creates a high supersaturation based on a small distance between source and seed and high temperature gradients, enhancing the polytype change towards 3C SiC. Additionally, the growth temperature is kept under 2000 °C in combination with an Si rich gas phase, which further stabilizes the cubic polytype. However, the nucleation of 3C SiC on hexagonal SiC polytypes is usually accompanied by the formation of double position boundaries (DPBs). During the nucleation of 3C SiC on hexagonal polytypes, the stacking sequence will be continued thermodynamically, dictated based on the underlying two bilayers [139]. Thus, if growth steps are present on the hexagonal substrate, 3C Seeds with varying stacking sequences will be formed on the different terraces, leading to defective merging during the lateral expansion of the 3C SiC, resulting in DPBs and SFs. Neudeck et al. [139] have presented a process to effectively eliminate the formation of DBPs during heteroepitaxial 3C SiC growth on 4H SiC substrates. They first produced a mesa-like structure by dry etching of (0001) orientated on-axis 4H SiC wafers. Afterwards, they used a homoepitaxial CVD growth process to completely remove atomistic growth steps from the mesas, followed by decreasing the temperature by 50–200 °C to initiate island growth of the cubic polytype. The lack of growth steps leads to the formation of 3C SiC seeds with the identical stacking sequence. Therefore, no DPBS will be formed during the merging of the 3C SiC seeds during lateral expansion. Note: The maximum size of the mesas was limited to 400×400 mm${}^{2}$ and DPB-free mesas can only be obtained if threading screw dislocations (TSD) are absent in the 4H substrate underneath the mesas. TSDs will act as an infinite source of growth steps, inevitably leading to the formation of stacking defects.

#### 4.2.3. Sublimation Growth

Different from the heteroepitaxial growth methods presented above, seeding material for the homoepitaxial growth of 3C SiC is not widely available yet. However, remarkable progress has been made on this topic in recent years. Schuh et al. [140] developed a transfer process to fabricate high-temperature stable 3C seeding stacks. Starting from 3C SiC grown heteroepitaxially on Si, the substrate was removed by wet chemical etching and the remaining free-standing 3C SiC was bonded to a polycrystalline SiC carrier using a carbon contain glue. By eliminating the low melting point of Si (1419 °C), the seeding stacks are suitable for the use in high-temperature sublimation growth processes, such as close space PVT (CS-PVT). Further information regarding the CS PVT setup can be found in [141] and chapter 5 of [142].

Using CS PVT and the mentioned seeding stacks, free-standing 3C SiC samples with diameters up to 100 mm and a thickness up to 3 mm could be realized [143,144]. The progress for the increase of sample diameter is depicted in Figure 11. Although the large area samples still suffer from cracking of the thin 3C SiC film during the removal of the CVD silicon substrate, the grown crystals show a high material quality, proved by the low full width half maximum (FWHM) of X-ray diffraction (XRD) measurements of the (002) reflex, which were as low as 140 arcsec. Note: Although a lot of research has been conducted to improve the material quality, the FWHM XRD rocking curve values for 3C SiC are still higher compared to 4H SiC where values in the range of 10 arcsec can be found in the literature [145,146]. Besides low FWHM values, it could be shown that the SF density as well as the overall stress in sublimation-grown 3C SiC crystal could be decreased compared to the heteroepitaxial-grown material on Si [143,147]. Based on the work of Anzalone et al. [124,125], which was already mentioned previously, free-standing, homoepitaxial-grown 3C SiC samples have become available. Although the material still has some setbacks, e.g., wafer bow, it is suitable as a starting point for the sublimation growth of 3C-SiC. Similar to the described seeding stacks, the homoepitaxial-grown seeds allow the growth of DBP-free material, as nucleating seeds will all have the same stacking sequence contrary to the heteroepitaxial growth on hexagonal SiC. However, compared to the seeding stacks, no transfer process is necessary for the homoepitaxial seeds, reducing the cracking probability and, consequently, increasing the up-scale potential. In fact, the first crack-free, free-standing large-scale 3C SiC crystals up to 650 µm thickness could be produced using this homoepitaxial seeding material, as can be seen in Figure 12 [148]. Further, Schöler et al. [149]) reported an overgrowth mechanism for protrusion defects using homoepitaxial seeds with an 4${}^{\circ }$ off-cut towards the [100] direction. Protrusions represent a setback that hinders the growth of bulk material up to several millimeters in thickness. This three-dimensional defect forms close to the interface between silicon and 3C SiC during heteroepitaxial growth on silicon and can therefore be viewed as intrinsic for the used seeding material. During further growth, this defect will latterly increase with increasing layer thickness and distort the material quality. Inevitably, the surface of grown crystal will be covered by protrusions at some point during the growth, depending on the defect density. Although efforts have been made to reduce the protrusion density [111,150], this problem is far from being solved, as complete elimination of the defect is required for real bulk growth with boules thicknesses up to 10 mm or more.

### 4.3. 4H Silicon Carbide

#### 4.3.1. Bulk Growth of 4H SiC

The growth of bulk 4H SiC is mainly carried out by utilizing the physical vapour transport (PVT) method. Demonstrated in 1978 by Tairov and Tsvetkov [151], PVT enabled large-diameter and gas-phase single crystalline growth of hexagonal SiC, and is employed to grow crystals of up to 200 mm in diameter (see Figure 13 and Figure 14). Other solution growth methods, such as top-seeded solution growth (TSSG) of SiC, are also conceivable. However, this method is held back by the low solubility of carbon in a silicon melt. According to the phase diagram, SiC exhibits a peritectic decomposition at approximately 2830 °C into carbon and an Si-rich Si-C solution [152]. It has to be mentioned that crystal growth from the melt is thermodynamically advantageous compared to vapour phase growth and very high-quality SiC crystals have been demonstrated with TSSG [153]. However, the obstacles of the non-stoichiometric melt composition, combined with the early availability and comparative ease of the implementation of PVT growth, led to an almost exclusive use of PVT for the production of commercial SiC substrates.

Crystal growth by PVT is taking place by inductively heating a graphite crucible to temperatures above 2000 °C. While the high temperatures of the PVT process allow good-quality crystals to grow, it also prevents any kind of CMOS compatibility. Inside the crucible, a SiC source, predominantly a powder, is placed below a single crystalline seed. A temperature gradient between the source powder and the seed ensures that mass transport will take place once sufficient growth temperatures are reached. The SiC powder source will sublimate and the resulting SiC gas species will recrystallize at the slightly colder seed. The mass transport, and therefore the growth rate, can be adjusted by varying the growth temperature in general, the axial temperature gradient inside the crucible or the growth pressure. All materials inside the growth chamber of the PVT setup need to be stable at high growth temperatures without breaking down in order to prevent changing growth conditions during the growth run or unintentional doping of the growing crystal. For this reason, graphite parts and isolations are utilized combined with a gas composition of argon and, if necessary, nitrogen for n-type doping. P-type doping with aluminium is possible as well, as demonstrated by Wellmann et al. [155]. The crystal growth is started by heating up to growth temperatures and the subsequent lowering of the pressure. The growth rate lies in the range of 50 to 500 µm/h but, depending on the growth cell design, can be set to rates up to 1 mm/h. Since hexagonal polytypes of SiC are grown at much higher temperatures than the cubic polytype, the crystal quality also tends to be of a higher quality with less structural defects. A more extensive description of the mechanics of PVT growth can be read elsewhere [156,157].

#### 4.3.2. Defects in Hexagonal SiC

The main challenge in PVT growth lies in the minimization of defects present in the crystal lattice. The diminishing effect of defects on the performance of SiC-based devices has been reported extensively. Micropipes (MP) can lead to breakdowns of p-n junctions [158,159], high densities of threading screw dislocations (TSD) have been correlated to lower breakdown voltages of 4H-SiC rectifiers and basal plane dislocations (BPD) are reported to increase the leakage current in MOSFETs [160] or JFETs [161]. While threading edge dislocations (TED) are also reported to reduce the breakdown voltage of Schottky devices [162], they have the least impact of the aforementioned defect types. Defects can either be generated due to process conditions or are inherited from the seed. MPs, TSDs and TEDs lie parallel to the growth direction while BPDs lie within the basal plane perpendicular to the growth direction. Dislocations can be revealed by selective etching with molten KOH at 500–520 °C for 5–10 min [163]. Only the Si-terminated face (so the (0001)-plane) is etched in an anisotropic fashion. The (0001)-plane or C-face is etched with an etching rate four times as fast and does not reveal any etch pits [164].

Figure 15 depicts a typical SiC wafer surface etched with molten KOH. All main defect types are shown. The BPD exhibits a characteristic elongated etch pit since the dislocation line lies perpendicular to the growth direction. BPDs are only visible in KOH-etched samples if there is a slight off-axis angle in respect to the (0001)-plane. TEDs are expressed by a small circular etch pit while TSD etch pits have a more pronounced hexagonal shape. In Figure 15, you can see several sizes of TSD etch pits. The reason for that lies in the different values the burgers vectors of TSDs can assume. It was demonstrated that TSDs exist with burgers vectors b ≤ 3c [165]. The different sizes of the TSDs in the depicted KOH-image most likely correlate to different amounts of b. The largest etch pits belong to MPs. Since MPs are essentially TSDs with an open core and burgers vectors b ≥ 3c [166], the etch pit looks up-scaled but similar. The overlap between MPs and TSDs at 3c means that, without directly observing the open core of MPs, it is very difficult to differentiate between a 3c MP and a 3c TSD in a KOH etching image. Other defect characterization methods worth mentioning are photoluminescent mapping, X-ray topography or Raman spectroscopy. A more extensive overview can be found in [167].

MPs are formed by carbon inclusions [168], adsorbing onto the growth interface during the process and inhibiting the step-flow, polytype switches during the growth [169] and insufficient backside protection against sublimation of the crystal [170,171]. These issues are largely resolved by now, demonstrated by growing MP-free 4H-SiC single crystals of up to 200 mm diameter [172]. The main focus currently lies in the reduction of the TSD and BPD density. TSDs are known to be formed due to the formation of independent growth islands during the initial seeding phase [173]. This can be remedied by utilizing off-axis seeds. BPDs, on the other hand, are induced by stress in the crystal, either during or after the growth is concluded, since deformation in hexagonal SiC polytypes takes place in the <112$\bar{0}$>{0001} slip system. As a conclusion, the stress present in the crystal and the BPD density is closely connected. To reduce the BPD density, the different sources of stress have to be considered. One is the difference in the coefficient of thermal expansion (CTE) of the different materials utilized in PVT growth. A crystal can be considered to be growing in a stress-free manner at growth temperatures since the thermal energy will enable it to include dislocations to adapt to present strain immediately. If the crystal is firmly connected to graphite parts, such as the seed holder or the crucible wall, during cool-down, the different CTEs of SiC and graphite will lead to stress and the generation of BPDs before reaching the ductile-brittle transition temperature of 1050 °C [174]. However, even if all contact to graphite is prevented, the temperature gradients inside the crystal before cooling down will induce stress once room temperature is reached [175,176]. This is also true for the radial temperature gradient. It was demonstrated numerically and experimentally that, from the shoulder region of a crystal, BPD arrays form and propagate into the crystal [177,178]. Therefore, low radial and axial gradients are preferred, without either inducing a concave growth interface or reducing the mass transport between source and seed too much. The control of the temperature gradients becomes even more important if bigger diameters of crystals, such as 200 mm, are to be grown since the total difference in temperature between the center of the crystal and its edge increases. Additionally, nitrogen doping in SiC modifies the CTE of hexagonal SiC [179]. Due to the nature of PVT growth, during the seeding phase, adsorbed nitrogen gas species can release from the graphite isolations, which subsequently leads to a sharp increase of doping for the first few µm of crystal growth. This, in turn, leads to a high amount of stress during the cool-down phase [180]. A controlled gradual increase of the nitrogen gas flux during the start of growth will prevent the inhomogeneous doping and therefore avoid excessive BPD formation. In addition, the utilized graphite parts should be of high purity, in the range of 6N, optimally combined with a purging step prior to growing. This is especially important if semi-insulating SiC for the novel photonic application in the topic of quantum information is desired, elevating the need for purity into magnitudes of 7N to 8N.

#### 4.3.3. X-ray Imaging

The high temperatures during the PVT process prevent a direct observation of the growing crystal. PVT setups are mostly limited to measure the temperatures above and below the growth cell with optical pyrometers through thin channels in the graphite isolation. However, it is possible to utilize 2D X-ray imaging to approximate the in situ growth rate and powder consumption [181,182,183], enabling the fine-tuning of parameters, such as the thermal field and consumption rate of the SiC source. For more specific inquiries of the growth process, advanced 3D imaging can be employed. Such sophisticated systems are not commercially viable due to the technical complexities; however, in an R&D environment, they have been proven to be incredibly valuable in considering problems such as the evolution of the crystal’s facet during growth [184], the specific morphology of the powder source for the determination of dynamic source material properties [185], as well as the impact of the curvature of the growth interface on the defect distribution in the resulting crystal [186].

#### 4.3.4. Numerical Modeling

Besides X-ray imaging, numerical modeling is utilized to further characterize the growth cell. This tool can be used to adjust a wide range of parameters, such as the shape of the temperature field inside the crucible, without having to perform time- and material-consuming test runs [185,187,188]. Figure 16 depicts a typical modeling result. To obtain a high accuracy of the modeling results, the main task is a precise knowledge of the material’s properties over a wide range of temperature. Laser flash analysis can be employed to obtain the temperature-dependent thermal conductivity while dilatometry is useful to characterize the thermal expansion. The temperature measurements provided by pyrometers can be used to validate the model. Further research of the SiC material system allowed to model the mass flow inside the crucible during a process [187,189,190]. In [190], ${}^{13}$C was utilized in the powder source, combined with X-ray imaging to track the mass flow during a PVT growth run and the results were subsequently compared with the numerical calculations. The growth kinetics within the growth cell and the stress acting on the growing crystal were investigated by several workgroups [180,191,192], followed by the assessment of the formation and movement of dislocations [136,177,193,194]. State-of-the-art modeling of the PVT growth cell allows the accurate depiction of the conditions present during the growth process, which would otherwise not be obtainable.

#### 4.3.5. CVD Growth of 4H-SiC

Like the cubic polytype, 4H-SiC can also be epitaxially grown employing the CVD method, as was first reported by Matsunami and Kimoto [195]. Since the 3C polytype is thermodynamically more stable than the hexagonal polytypes at the typical growth temperatures of the CVD process, homoepitaxial growth of 4H-SiC needs off-cut substrates to prevent polytype switches. As long as the step-flow growth mode, as described by the BCF model, is retained [196], the polytype of the substrate can be reproduced even though the thermodynamic conditions for hexagonal SiC are unfavorable at lower temperatures [197]. This holds true as long as there are no growth islands on large-area terraces. One important advantage of the CVD-growth of 4H-SiC layers is the conversion of BPDs to TEDs through image-force effects present at lower off-cut angles, such as 2 to 4${}^{\circ }$ [198,199]. To some degree, this conversion is also happening in PVT-growth, although there, it is much less controllable. The general mechanics of homoepitaxial CVD growth are the same as for the heteroepitaxial growth of 3C on Si, described in the previous section. Special care has to be taken to prevent disruptions of the step-flow caused by side reactions during the growth process, such as condensation of Si atoms and resulting Si droplets. While a lower off-cut angle of the substrate increases the image-force and, therefore, the conversion of BPDs into TEDs, it also increases the size of terraces and the probability of 3C-SiC islands. To prevent this, the diffusion length of an adsorbed Si or C atom on the substrate should at least reach half of the terrace width. The main parameters to vary are the growth temperature and the gas fluxes of the precursor gases silane and propane. Compared to PVT-growth, the growth rate seldom reaches values above 50 µm/h, making CVD-growth unsuitable for bulk growth of SiC. However, the process temperature can be set to values below 1200 °C, making a CMOS compatibility possible.

## 5. Conclusion and Perspectives

The exceptional optical properties of bulk SiC will be enhanced when light is confined in nanoscale waveguides, so that the devices become more compact and energy efficient. Broadly speaking, SiC has been used in the photonic field in such cases as being a substrate for high-efficiency GaN-based LED because of its small lattice mismatch to GaN, emitting light by doping donar-acceptor pairs and forming porous structures [200,201]. In this review paper, a novel photonic application of SiC is presented. Although this field is very young, about 10 years old, many breakthroughs have been achieved, thanks to the already mature SiC growth and fabrication technologies. Compared to the longer-existing material platforms for photonic integrated circuits (Si, SiO${}_{2}$, SiN, LiNbO${}_{3}$, AlGaAs, etc.), SiC is outperforming on both its nonlinear optical properties and its potential as a perfect candidate for a single photon source for future quantum photonic integrated circuits. SiCOI stacks with a polytype of 4H, 3C and amorphous have been demonstrated by different formation methods, and high-quality-factor (>1 million) microring resonators have been reported. A lot of passive and active devices are demonstrated, such as beam splitter, polarization beam splitter, modulator, frequency comb, etc [23,38,63,74].

From a loss reduction point of view, any defects causing absorption and scattering of light at the device working wavelength should be eliminated. However, there are a bunch of point defects in SiC which are optically addressable and could be potentially used for a single photon source, a key device for quantum technology. Integrating SiC single photon source, monolithically, with other building blocks enables SiCOI a game-changer for the future of quantum PIC [202].

But, before SiC really plays a crucial role in QPIC, further research and development are needed:(1)Low-loss SiCOI stacks are widely available.(2)The insertion loss of the chip, including propagation loss and coupling should be less than 3 dB.(3)More SiC-based quantum devices should be demonstrated, such as quantum memory, detectors, etc.(4)SiC color centers should be further explored and prepared as a competitive candidate for a single photon source for real applications in communication, computing, etc.

## 6. Impacts of Novel Photonic Applications of SiC

The research in silicon carbide started to have strong momentum in the 1990s. That decade had a heavy focus in research and in Europe several EU projects were initiated. The first ones were mainly in bulk growth and included some epitaxy and device development. Epitaxy and device challenges were heavily related to defects in research-grade wafers, and epitaxy recipes and issues such as stable electrical contacts were not in place. Test devices suffered from structural variations within wafer areas, as well as between wafers. The first European wafer suppliers emerged 20–25 years ago and slowly substrates became more available for device development. The market started to grow slowly, firstly by the sale of materials. It was only when devices became available that the market started to grow faster. The SiC diodes and power electronics pushed the market, and also started to require more wafers. As a consequence, the wafer prices have steadily decreased and volume manufacturing is now established. Today, many market players are entering and the SiC device market will grow to billions of US dollars annually. The journey from research to a strong, growing SiC market has taken about 30 years.

The efforts in the research decade created a strong momentum in Europe. Researchers built their competence and gained network and personal relations. While conducting present projects, new ideas emerged through the exchange between researchers. The fluorescent silicon carbide for a new white LED was one of the European avenues which gathered a cluster of research groups in the year 2010, with teams from Denmark, Germany and Sweden, which now present this review article. They have continued to collaborate and propose new projects with other partners having complementary expertise. In such a way, silicon carbide research expanded to other avenues and started to grow to applications other than the original ones that were related to power devices. The European efforts in SiC from the 1990s have grown momentum.

The next phase of momentum is given by impact creation. Projects are typically described by complementary partners and value chains. The motivations are given by technical advantages and better performances. Increasingly, there are aims like the Green Deal, Sustainable Development Goals (SDGs), and others which have a long-term aspect. Even though those are reasonable motivations, there should be sooner goals which show synergies. This is where regional development of Smart Specialisation Strategies (S3) comes in. It is a place-based innovation policy concept in which regional priorities are supported. This is achieved through the entrepreneurial discovery process in innovative sectors, fields or technologies. It applies a bottom-up approach to find and support regional scientific and technological development. The S3 is, in simplified terms, described as ways to create more economical growth in European regions. By finding areas of strength and specializing in more efforts such as this one, there will be a sustainable way to increase economic growth. This is basically achieved by finding partners in regions and collaboration in regions which are complementary. It, in fact, describes various ways to have value chains. Thus, value chains are common in both technology and smart specialization motivations and implementation. The S3 was introduced about ten years ago. To date, more than 120 Smart Specialization Strategies are developed in EU regions and Member States. Now comes Sustainable Smart Specialization Strategies (S4). The S4 has an emphasis on the importance of taking a long-term view of the development of a region. It should not just be as an adaptive path, rather, it should be a proactive path.

The SiComb project is an H2020 EU-FET Open project that explores the photonic application using SiC, as described in the technical part of this paper. The EU FET Open, and EIC Pathfinder, as denoted in the Horizon Europe Programme, are projects that present ideas with potential radical technology leaps, looking forward 10–20 years. These are early stages of research with potential for high gain, if the research and technology challenges can be managed.

Clearly, there can be synergies and proactive development by aligning technology and regional development and their motivations to gather stakeholders. Creating exchange between different stakeholders, such as researchers, regional development actors, policy makers, citizens, and other relevant players will produce the momentum which aligns the development of smart specialization and technology for society. The common overall aim of sustainable approaches for SDGs and the Green Deal will be more efficient by considering such joint impact creation.

## Figures and Tables

**Figure 1 materials-16-01014-f001:**
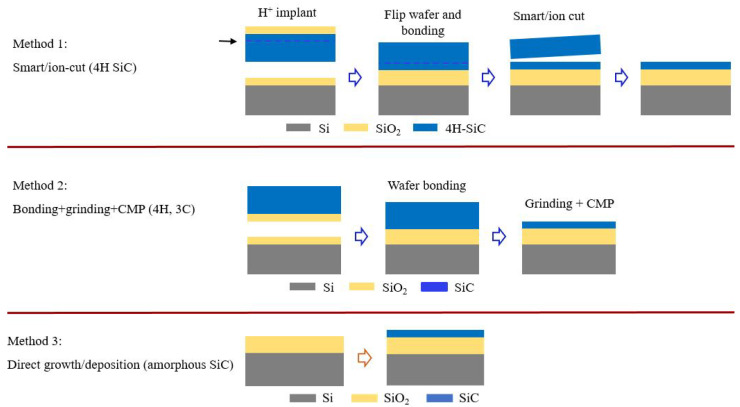
Three SiCOI stack formation methods.

**Figure 2 materials-16-01014-f002:**
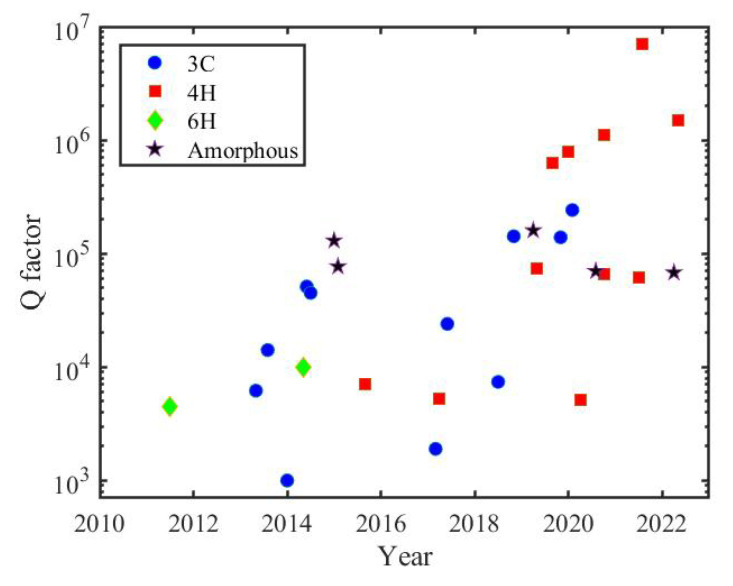
State of the art of SiC optical cavities [23,25,28,32,34,35,36,38,39,40,41,42,43,44,45,46,47,48,49,50,51,52,53,54,55,56,57,58].

**Figure 3 materials-16-01014-f003:**
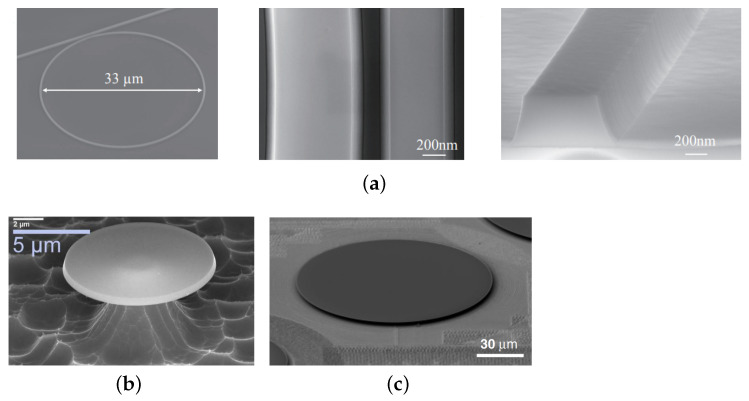
Scanning electron microscope images of (**a**) 4H SiCOI MMR, reprinted with permission from [36] ©The Optical Society, (**b**) 3C SiC MDR [59], and (**c**) 4H SiCOI MDR [56].

**Figure 4 materials-16-01014-f004:**
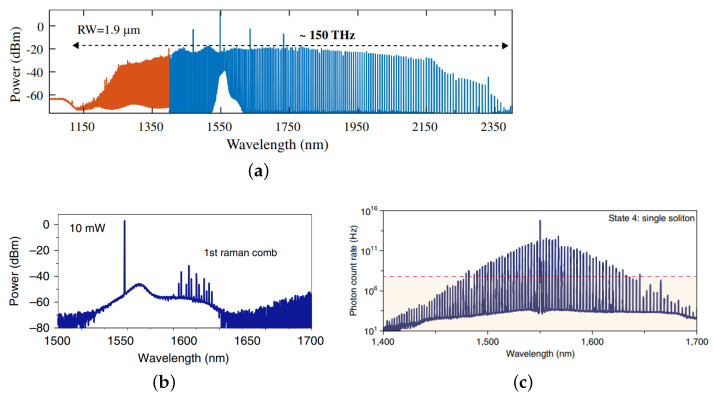
Experimental spectra of (**a**) octave-spanning Kerr frequency comb reprinted with permission from [57], (**b**) Raman comb, reprinted with permission from [56], and (**c**) quantum soliton comb, reprinted with permission from [71] in SiC microresonators.

**Figure 5 materials-16-01014-f005:**
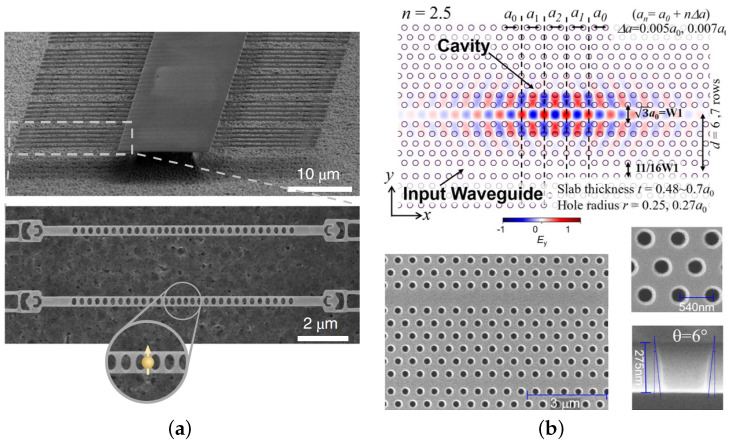
Scanning electron microscope images of (**a**) 1D photonic crystal nanobeam cavities, reprinted with permission from [52] and (**b**) 2D photonic crystal heterostructure cavity, reprinted with permission from [50] ©The Optical Society.

**Figure 6 materials-16-01014-f006:**
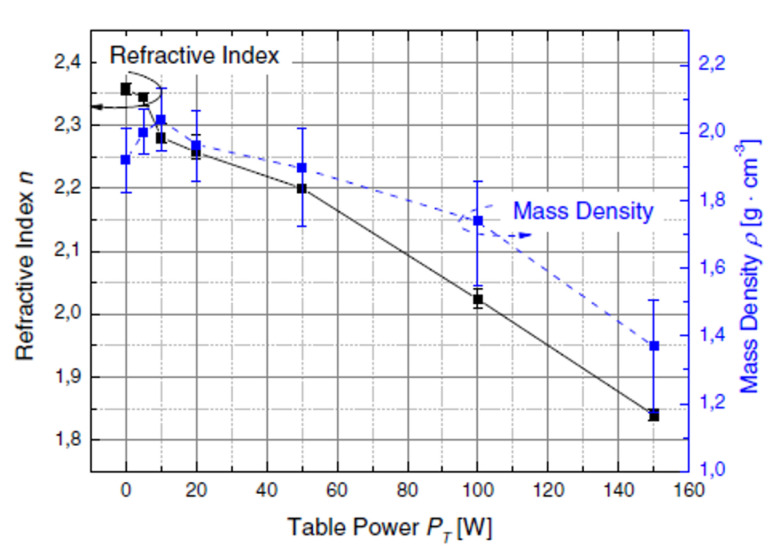
Evolution of the refractive index (left) and the density (right) as a function of the power applied on the substrate $P_{T}$ measured by Frischmuth team [83]. Reprinted with permission from Elsevier.

**Figure 7 materials-16-01014-f007:**
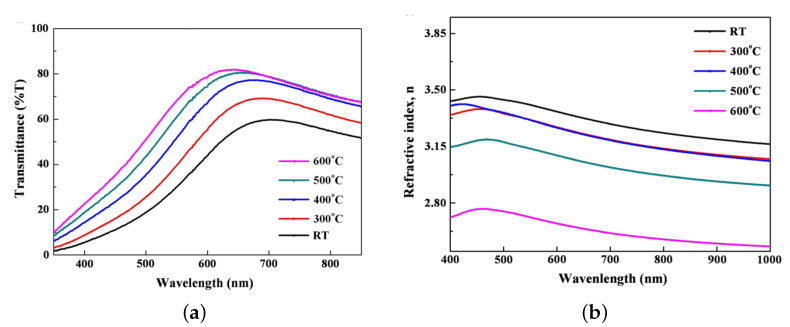
Evolution of (**a**) the transmittance and (**b**) the refractive index as a function of the temperature of the substrate, after Seo team [104]. Reprinted with permission from Elsevier.

**Figure 8 materials-16-01014-f008:**
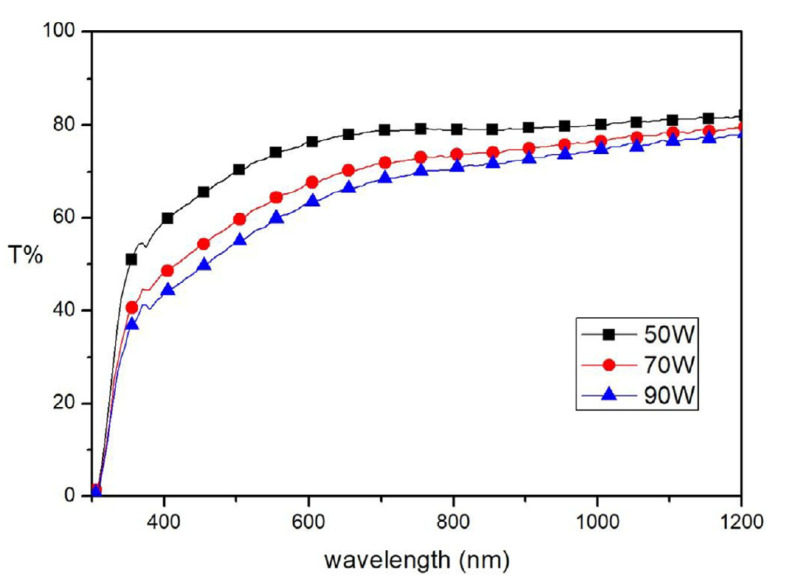
Evolution of the transmittance of SiC films in function of the RF power, measured by Wang [105]. Reprinted with permission from Elsevier.

**Figure 9 materials-16-01014-f009:**
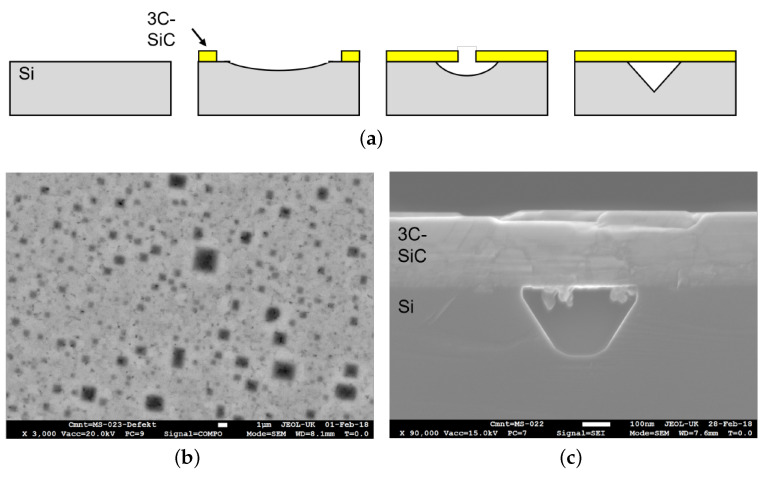
(**a**) Scheme of the formation of voids at the 3C-SiC/Si interface. (**b**) Top view scanning electron microscopy (SEM) image of a 3C on Si sample grown at 1200 °C. The darker areas are attributed to voids underneath the 3C-SiC layer. (**c**) Crosscut SEM image of a void at the Interface 3C-SiC/Si.

**Figure 10 materials-16-01014-f010:**
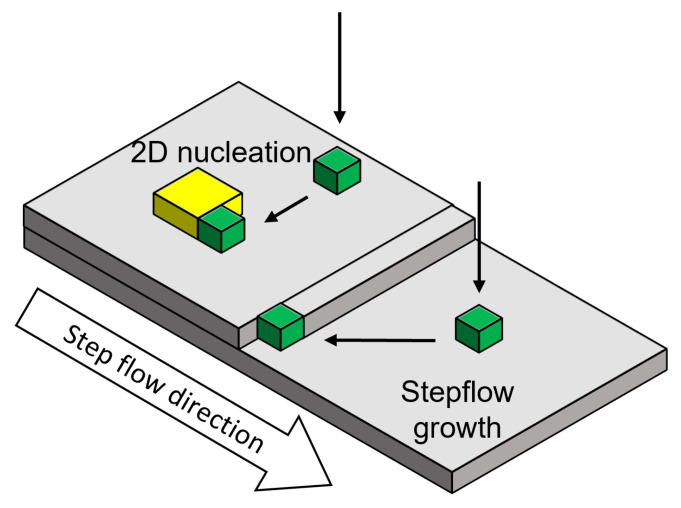
Scheme of two different growth schemes for the epitaxial growth. On the lower terrace, migration of the adatom at the edge of a growth step occurs, reproducing the substrate polytype. On the upper terrace, 2D nucleation takes place, which can lead to a switch in polytype.

**Figure 11 materials-16-01014-f011:**
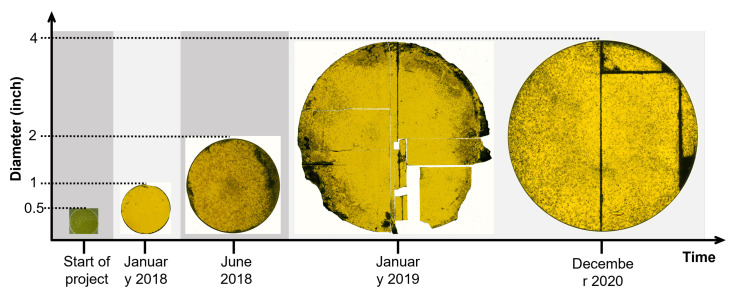
Evolution of diameters for bulk 3C-SiC crystals grown by sublimation growth during CHALLENGE project with indicated timeline. With permission from [145].

**Figure 12 materials-16-01014-f012:**
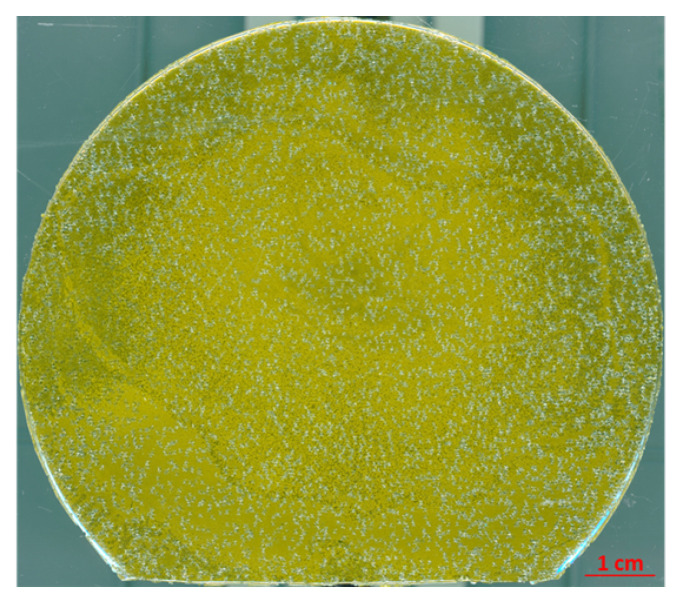
High resolution scan of free-standing 3C-SiC crystal grown by CS PVT on homoepitaxial seeding layer with a diameter of 92 mm and a thickness of 650 µm. With permission from [148].

**Figure 13 materials-16-01014-f013:**
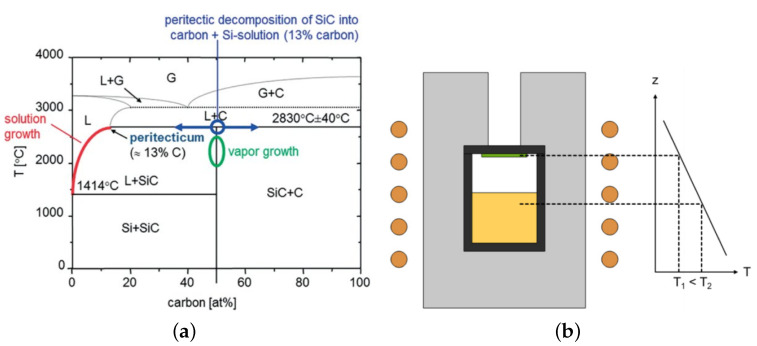
(**a**) Phase diagram of Si and C. (**b**) Schematic setup of a typical PVT growth cell and isolation. With permission from [154].

**Figure 14 materials-16-01014-f014:**
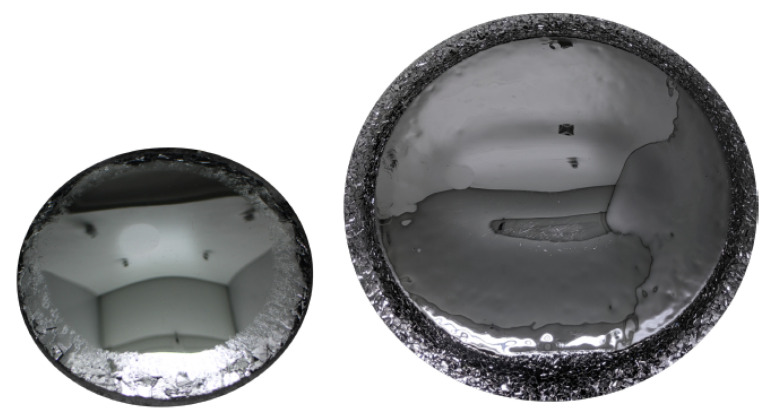
100 mm (**left**) and 150 mm (**right**) 4H-SiC crystal.

**Figure 15 materials-16-01014-f015:**
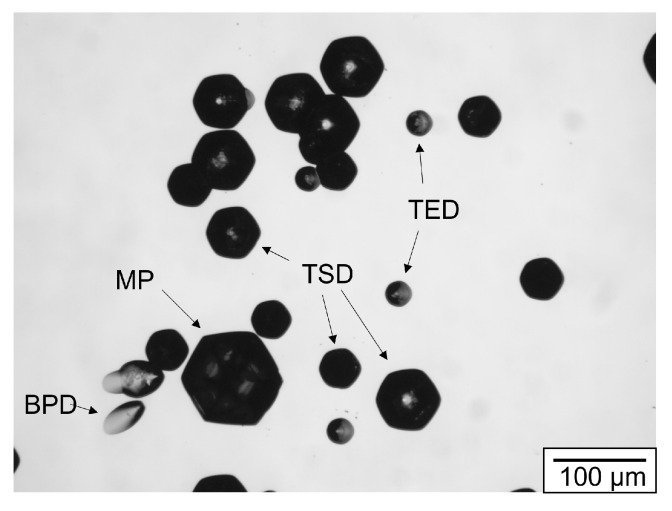
Microscopic image of typical etch pits in a 4H/6H-SiC wafer, etched with KOH.

**Figure 16 materials-16-01014-f016:**
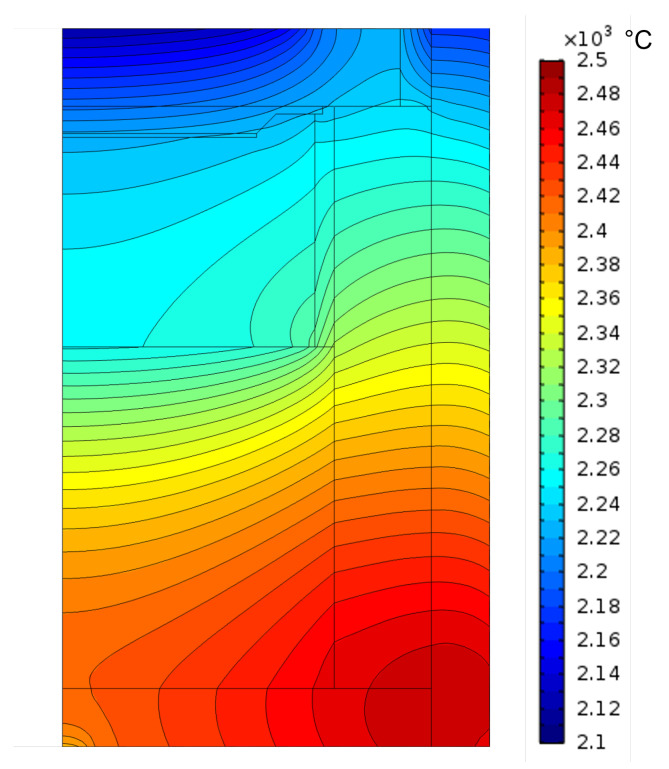
Calculated temperature field of a typical growth cell in a PVT reactor.

**Table 1 materials-16-01014-t001:** Main material platforms for PICs. *n*: refractive index, BG: bandgap, $n_{2}$: nonlinear refractive index, $\chi^{\left(2\right)}$: second-order susceptibility.

Materials	Key Features				Pros	Cons
	*n*	*BG* (eV)	*n*_2_ (m^2^/W)	*χ*^(2)^ (pm/V)		
Si [1,2,3,4,5]	3.48	1.12	4.5×10${}^{-18}$		High refractive index, compact, CMOS compatible, cost-effective	Small bandgap, strong two photon absorption
SiO${}_{2}$ [6,7]	1.5	9	3 × 10${}^{-20}$		CMOS incompatible, low loss	Low refractive index, low nonlinearity
SiN [8,9,10]	2.0	5	2.5 × 10${}^{-19}$		CMOS compatible, cost-effective	Low nonlinearity
Hydex [11]	1.7	1.8	1.2 × 10${}^{-19}$		CMOS compatible, cost-effective	Low nonlinearity
GaP [12,13,14]	3.1	2.26	1.2 × 10${}^{-17}$	200	High refractive index, high nonlinearity	Cost-ineffective, CMOS incompatible
AlN [15,16]	2.1	6	2.3 × 10${}^{-19}$	1	Efficient light source	Cost-ineffective, CMOS incompatible
Diamond [17]	2.4	5.5	8.2 × 10${}^{-20}$		CMOS compatible, wide bandgap	Cost-ineffective, difficult to etch
AlGaAs [18,19]	3.3	1.4–2	2.6 × 10${}^{-17}$	100	High refractive index, high nonlinearity	Small bandgap, cost-ineffective, CMOS incompatible
LiNbO${}_{3}$ [20,21,22]	2.2	4		27	Well-known optical properties, cost-effective	CMOS incompatible, difficult to etch
SiC [23]	2.6	2.4–3.2	1.3 × 10${}^{-18}$	33	High refractive index, compact, CMOS compatible, high nonlinearity, light emitter	Cost-ineffective

**Table 2 materials-16-01014-t002:** State of the art of SiC waveguides.

Material	Stack	Method	Size (nm${}^{2}$)	Loss (dB/cm)	*Q*	n${}_{2}$ (m${}^{2}$/W)
3C SiC [33]	SiC-Si	Floating	730 × 500			5.3 × 10${}^{-19}$
3C SiC [34]	SiCOI	2	1700 × 500	2.9	1.42 × 10${}^{5}$	
a SiC [35]	SiCOI	3	800 × 350	3	1.6 × 10${}^{5}$	4.8 × 10${}^{-18}$
4H SiC [36]	SiCOI	1	750 × 500		7.3 × 10${}^{4}$	
4H SiC [37]	SiCOI	1	2700 × 600			8.6 × 10${}^{-19}$
4H SiC [38]	SiCOI	2	3000 × 530	0.38	1.1 × 10${}^{6}$	6.9 × 10${}^{-19}$

## Data Availability

The data presented in this study are available on request from the corresponding author.

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
