# Peer review of "Novel Photonic Applications of Silicon Carbide"

_materials, 2023, doi:10.3390/ma16031014_

Round 1

Reviewer 1 Report

Please see the enclosed file.

Author Response

Response to Reviewer 1

This article reviews the optical properties of SiC, its fabrication, and its characterization. The content of the review is very useful. However, authors should make an effort to make it attractive for a wide audience. Only those with knowledge of SiC-related applications and the pabrication process of SiC-based components will find this version useful. Also, please address the following issues:

Thanks a lot for the very positive comment. This review paper is already very comprehensive covering widely from material growth and characterization, thin film transfer, optical devices nanofabrication and characterization, as well as their impact. Because of this, four corresponding authors with complementary expertises are listed in order to address the post queries from wide audience.

Comments:

  1. Keep all the variables in italics.

In abstract, define the abbreviation Q. Do so throughout the article, even though they are widely used and known to the readers.

In abstract, all the abbreviations have been defined, including SiC, SiCOI, CMOS, Q. The variables in the manuscript have been changed into italics.

  1. In the following, "enhancement" is not an appropriate word; I think the word "amplification" suits better here.

The light enhancement in the microresonators can be reflected by the photon density.

Instead of using light enhancement or amplification, we decide to use field enhancement, which is a more commonly recognized term.

  1. Provide reference to equation (2)

Response: Thanks for this comment. We have added a reference of the Purcell factor expression: “Pelton, M. Modified spontaneous emission in nanophotonic structures. Nature Photonics 2015, 9, 427–435.”, and replaced the original equation with the equation mentioned in this referenced paper.

  1. Line 146, authors wrote:

“….and the enhancement magnitude is expressed by Purcell factor in Equation (2).”

Response: We have modified the sentence by “Because of the potential of emitting single photons by defects in SiC, photonic crystal cavities are studied to enhance the emission rate of the defects by addressing the defects in the cavities. The amplified magnitude of modified emission rate comparing with the defects in free space is expressed by Purcell factor in Equation 2.”

  1. Please specifically mention that F is the Purcell factor (i.e., Purcell factor (F)).

Response: Thanks for this comment. We have added the detailed expression of the mentioned abbreviations in the Equation 2.

  1. Authors wrote:

“In order to realize large enhancement, the quality factor of the photonic crystal cavity 149 should be as large as possible, and the mode volume is preferred to be kept small.”

Please specifically mention what is meant by “large enhancement”. Please mention enhancement in what parameter.

Response: We have revised the related sentences by “In order to realize large increase of emission rate of photons from defects in SiC, the stronger light-matter interaction is required. From Equation 2, the quality factor of the photonic crystal cavity should be as large as possible, and the mode volume is preferred to be kept small.”

  1. Authors wrote:

The refractive index decreases with the density, which can be tuned by varying the power

Please mention varying the power of what?

Response: the sentence has been corrected as follows: “The refractive index decreases with the density, which can be tuned by varying the plasma power”

  1. Please write the conclusion at the end of the review.

Conclusion has been merged into the perspective section.

  1. Make the language more academic. I noticed many incomplete sentences. This is very important. Otherwise, readers may lose interest in reading the article.

Proofreading has been done by authors one more time.

  1. It would be nice if the authors could provide a list of substrates and their thermal expansion coefficient values.

Thanks for the suggestion. Although thermal expansion coefficient is an important property for bonding processes between two different materials, it is not a crucial optical property, and is rarely investigated in SiC integrated photonic platforms. Thus, after consideration, we decide not to introduce and compare the thermal coefficients of different materials.

Reviewer 2 Report

after reading the reply of the authors, I recommend to accept the manuscript

the authors mention to LiNbO3 crystal as a potential Nonlinear materials, can the authors add more about LiTaO3 materials it is just like LiNbO3 crystal expect replacing Nb wit Ta atoms.

Author Response

Thanks for bringing up a new material of LiTaO3. LiNbO3 is regaining interest in nonlinear optics because of the wide availability of commercial LNOI stacks which is not the same as LiTaO3 even if they owe the similar bulk optical properties.

Reviewer 3 Report

The manuscript “Novel Photonic Applications of Silicon Carbide” present a detailed review on the photonic applications of SiC material. The work provides latest developments in the SiC based photonic devices that may be useful to scientific community working in this area. The manuscript is well written and included relevant literature. Based on above, the work may be accepted for publication following a minor revision as given below:  

1.      In section 2: Silicon carbide photonic integrated platforms—Can authors provide some more details on which method (i.e., 2 or 3) is superior than other and why?

2.      Can authors provide some details on why 6H is not so popular in making optical cavities.

3.      Figure 5 (a), Can authors add scale in figure (a).

4.      Authors have explained the SiC deposition/growth techniques in much detail that are good. However, their effect on the targeted application, i.e., photonic devices, is vague. Therefore, authors are encouraged to include such details in the manuscript.

Author Response

Response to Reviewer 3

The manuscript “Novel Photonic Applications of Silicon Carbide” present a detailed review on the photonic applications of SiC material. The work provides latest developments in the SiC based photonic devices that may be useful to scientific community working in this area. The manuscript is well written and included relevant literature. Based on above, the work may be accepted for publication following a minor revision as given below: 

  1. In section 2: Silicon carbide photonic integrated platforms—Can authors provide some more details on which method (i.e., 2 or 3) is superior than other and why?

A paragraph has been added at the end of section 2 for further comparison of 3 different methods.

  1. Can authors provide some details on why 6H is not so popular in making optical cavities.

A short paragraph is added after the method comparison at the end of section 2.

  1. Figure 5 (a), Can authors add scale in figure (a).

Two scales have been added to Figure 5 (a).

  1. Authors have explained the SiC deposition/growth techniques in much detail that are good. However, their effect on the targeted application, i.e., photonic devices, is vague. Therefore, authors are encouraged to include such details in the manuscript.

In order to clarify the motivation for presenting the current state-of-the-art of processing the various SiC materials, the following text has been added at the beginning of Section 4:

“The most promising large area thin film deposition processes (=amorphous and micro-crystalline material) are of particular interest due to their huge variability in terms of stacking thin optical films on almost any kind of substrate and because of their comparably low manufacturing costs. The currently mid-range optical loss may be reduced by further improvement of the deposition processes. Therefore, in the following an overview on all SiC-based thin film methods currently being under development will be given. The optical lowest loss may be achieved in single-crystalline SiC material exhibiting an as low as possible dislocation density and unintentional doping. At the same time, by adaption of the single-crystal growth process intrinsic point defects may be generated for single photon source or qubit applications. Besides the preparation of high-quality 4H-SiC crystals and wafers, also bulk growth of 3C-SiC will be outlined. The latter material exhibits a higher defect density than 4H-SiC. However, its higher crystalline symmetry may be of advantage at a certain stage if optical active centers (= point defects and intentional doping) are added as functionality to the electrical-optical devices.”

In addition it is noted that the part on amorphous –SiC is presented with the optical properties point of view (examples of Fig 6, 7, 8).